# Models of Aviation Noise Impact in the Context of Operation Decrease at Tan Son Nhat Airport [note 1]

**DOI:** 10.3390/ijerph20085450

**Published:** 2023-04-10

**Authors:** Tran Thi Hong Nhung Nguyen, Bach Lien Trieu, Thu Lan Nguyen, Makoto Morinaga, Yasuhiro Hiraguri, Takashi Morihara, Yosiaki Sasazawa, Tri Quang Hung Nguyen, Takashi Yano

**Affiliations:** 1Graduate School of Natural Science and Technology, Shimane University, Matsue 690-8504, Japan; 2Department of Architecture and Building Engineering, Faculty of Architecture and Building Engineering, Kanagawa University, Yokohama 221-8686, Japan; 3Department of Architecture, Kindai University, 3-4-1 Kowakae, Higashiosaka 577-8502, Japan; 4Department of Architecture, National Institute of Technology, Ishikawa College, Kitachujo, Tsubata 929-0392, Japan; 5Faculty of Education, University of the Ryukyus, 1 Senbaru, Nakagami, Nishihara, Okinawa 903-0213, Japan; 6Faculty of Environment and Resources, Nong Lam University, 6, Linh Trung, Thu Duc, Ho Chi Minh City 700000, Vietnam; 7Graduate School of Science and Technology, Kumamoto University, 2-39-1 Kurokami, Chuo-ku, Kumamoto 860-8555, Japan

**Keywords:** structural equation model, noise impact, health risk, community response, annoyance, sleep effects, insomnia

## Abstract

Air traffic bans in response to the spread of the coronavirus have changed the sound situation of urban areas around airports. This study aimed to investigate the effect of this unprecedented event on the community response to noise before and after the international flight operation at Tan Son Nhat Airport (TSN) in March 2020. The “before” survey was conducted in August 2019, and the two “after” surveys were conducted in June and September 2020. Structural equation models (SEMs) for noise annoyance and insomnia were developed by linking the questionnaire items of the social surveys. The first effort aimed to achieve a common model of noise annoyance and insomnia, corresponding to the situation before and after the change, respectively. Approximately, 1200 responses were obtained from surveys conducted in 12 residential areas around TSN in 2019 and 2020. The average daily flight numbers observed in August 2019 during the two surveys conducted in 2020 were 728, 413, and 299, respectively. The sound pressure levels of the 12 sites around TSN decreased from 45–81 dB (mean = 64, SD = 9.8) in 2019 to 41–76 dB (mean = 60, SD = 9.8) and 41–73 dB (mean = 59, SD = 9.3) in June and September 2020, respectively. The SEM indicated that the residents’ health was related to increased annoyance and insomnia.

## 1. Introduction

Environmental noise is a growing concern worldwide due to its significant impact on mental health and well-being. The accumulated data from socio-acoustic surveys have provided a scientific basis for the identification of the threshold of noise exposure that poses a risk to human health. The document “Guidelines for Community Noise” was released by the World Health Organization (WHO) in 1999; it determines the limit levels for each noise source [1]. In a systematic review of environmental noise and annoyance to support the development of the WHO guidelines [2,3], exposure to environmental noise was found to lead to annoyance and sleep disturbance and to have negative effects on health and well-being. The Environmental Noise Guidelines for the European Region, launched in 2018, introduced stricter limit values for aircraft noise based on the existing evidence related to the health impacts on people living around airports [4].

Another WHO report on the burden of disease resulting from environmental noise, quantified by the number of years of healthy life lost due to exposure to environmental noise in Europe, found that environmental noise was a significant environmental health risk and that reducing noise exposure could improve public health [5]. In a cross-sectional study in six European countries which investigated the association between exposure to aircraft and road traffic noise and heart disease and stroke, the exposure to both types of noise was found to be associated with a higher risk of cardiovascular disease [6]. In a review of the epidemiological evidence supporting the role of environmental stressors, including noise and air pollution, in the development of cardiometabolic diseases, strong evidence linking noise exposure to hypertension and cardiovascular diseases was confirmed [7]. This suggests that mitigation strategies, such as noise reduction and green space development, can have a positive impact on health. For example, the effect of a quiet façade on the annoyance caused by urban road traffic noise is greater for people who are more sensitive to noise [8]. Overall, these studies emphasize the need for better noise exposure assessment and more research on the health effects of noise decrease interventions.

Although the 1999 guidelines were primarily based on European and North American surveys, the 2018 guidelines cite data from Asia, enabling national governments worldwide to develop noise limits or standards and to decide on noise-reduction measures. However, acoustic quantities can only explain the proportion of variance observed in the community response to noise. The annoyance and disturbance responses were found to vary significantly in response to contextual effects such as residential conditions, neighborhood environment, demographic variables, and personal factors [9,10]. A review of the research progress on the community response to noise from 2017 to 2021 reported significant variance in the level of annoyance at a given sound level [11]. Therefore, general exposure–response functions cannot reflect the local situation. For local noise management, exposure–response information tailored to the situation in local communities would be more helpful for noise interventions that aim to minimize noise health effects, including annoyance. Therefore, an analysis of the data of socio-acoustic surveys that takes into account the associations between the acoustic and non-acoustic factors and the health consequences and their contribution to the effect of noise intervention on the local community response is necessary.

There is considerable variability in the annoyance within specific noise exposure groups, indicating that other factors, such as personal characteristics, may play a role. A review of the studies on the effects of noise on health, including both auditory and non-auditory effects, found that noise exposure is associated with a range of health outcomes, including annoyance, sleep disturbance, cardiovascular disease, and cognitive impairment [12]. Schütte et al. (2014) used structural equation modeling to analyze the annoyance caused by environmental noise and found that personal factors such as noise sensitivity, age, and education level, as well as the physical characteristics of the noise itself, had a significant effect on annoyance [13]. Overall, these studies suggest that noise exposure can have negative health effects, including annoyance and sleep disturbance, and that personal factors may play a role in determining the extent to which noise exposure affects individuals.

The Tân Sơn Nhất (TSN) international airport is the busiest airport in Vietnam [14]. Before the epidemic outbreak, the number of flights operating at the TSN airport had been increasing continuously over the years to meet the growing demand for air travel. Noise exposure around the TSN airport increased from 53 to 71 dB *L*_den_ (day–evening–night weighted sound pressure level) in 2008 to 63 to 81 dB in 2019 [15]. Extremely high levels existed in almost all areas in the vicinity of TSN, inside a dense residential area of the most active metropolitan area in Vietnam. However, after the epidemic outbreak in early 2020, Vietnam blocked some international flights from and to TSN in January and completely shut down its borders in March 2020. The significant change in the acoustic environment around TSN owing to this event enabled a study that could compare community responses before and after the change.

Research on community reactions in the context of noise reduction has been conducted in developed countries; however, there are few precedents in developing countries. Furthermore, few studies have examined changes in the noise exposure levels owing to changes in an airport’s operational conditions [16,17,18]. This study aimed to provide tailored exposure–response information to minimize noise health effects in the context of the operation decrease at the TSN airport by analyzing socio-acoustic survey data and investigating the associations between acoustic and non-acoustic factors and their contribution to noise interventions aimed at local community responses. The preliminary findings were reported at the Internoise conference [19]. In this paper, we will present detailed results, as well as a more comprehensive analysis and discussion. The causal structures of noise annoyance and insomnia in the 2019 and 2020 surveys were compared to determine whether community health differed under changes in the acoustic environment.

## 2. Materials and Methods

### 2.1. Survey Plan

Socio-acoustic surveys were conducted at five sites under the aircraft landing paths on the east side (Sites 1–5), five sites under the takeoff paths on the west side (Sites 6–10), and two control sites to the north of the airport (Sites 11 and 12) (Figure 1). The surveys were conducted via face-to-face interviews during the day on weekends. We recruited approximately 50 students from Nong Lam University, Ho Chi Minh City, as interviewers. The students were trained in implementing the social survey before participating in the survey. The interviewers visited approximately 100 households at each selected survey site and interviewed one adult from each family during the first survey in August 2019. The composition of the interviewees in each household was adjusted to have the same rate of demographic factors as those in the Vietnam Census. To ensure a balance between males and females and between different generations, fathers, mothers, and other adults in the family were selected for the survey.

This investigation was conducted seven months before the change owing to the complete cessation of international flight operations implemented in March 2020, when TSN was operating at its highest capacity. The respondents from the 1st survey were revisited for the 2nd survey, three months after the change. In the 3rd survey, the investigation was conducted six months after the change in the same area as the 1st and 2nd surveys. The respondents of the 3rd survey differed from those who participated in the 1st and 2nd surveys.

### 2.2. Questionnaire and Measuring Scales

Annoyance and sleep effects are significant effects of noise on the community. In this study, although many different health effects were included in the survey, the analysis and modeling were conducted with a focus on these two noise health effects. The questionnaire was compiled based on the questions used in previous surveys at the Noi Bai Airport [20]; these questions were statistically analyzed, and the reliability and validity of the questions were examined. We added health-related questions that were used in surveys on the health effects of aircraft noise; the questions were translated from English and Japanese. The questionnaire was checked for plausibility in terms of language and length by the local students who participated as interviewers. The questionnaire items included the technical specification ISO/TS 15666 [21], the Total Health Index (THI) [22], the Center for Epidemiologic Studies Depression Scale Revised (CESD-R10) [23], the Kadena study on insomnia and hypertension questionnaire [24], and questions to identify insomnia and hearing loss [25,26]. The items concerning the change in contextual factors and the interactive effects of acoustic and non-acoustic variables included convenience and preference for residential areas, activity interventions, and health-related personal information.

The annoyance effect was measured as the percentage of respondents who were highly annoyed (%HA). %HA is the percentage of respondents who chose 8, 9, or 10 from an 11-point numerical scale (0 to 10). Similarly, sleep effect was measured as the percentage of respondents who suffered from insomnia. A questionnaire on insomnia was created by referring to the Insomnia Symptom Questionnaire (ISQ). Accordingly, the percentage of respondents with insomnia (%ISM) referred to those who responded affirmatively to “have any trouble with sleep,” “sleepy during daytime and cannot work well more than three times a week,” and had experienced at least one of the other symptoms (1)–(4) listed in the ISQ more than three times in a week. Table 1 lists the questions and scales used in all the surveys to evaluate annoyance and sleep effects.

### 2.3. Aircraft Noise Estimation

In the 1st survey, aircraft noise exposure, day–evening–night noise levels (*L*_den_), and nighttime noise levels (*L*_night_) were measured and estimated using an Integrated Noise Model 7.0 (INM). The INM is a computer model used to predict and analyze aircraft noise exposure around airports [27]. It is designed to estimate the noise levels generated by aircraft engines and takes into consideration the types of aircraft, flight patterns, and operating conditions. The flight log data of the entire period were provided by the airport office. The flight path data were collected from the Automatic Dependent Surveillance-Broadcast receiver installed in the airport management office. The model used A-weighting, the most common weighting system for assessing the effect of aircraft noise on the community, to adjust for the sensitivity of the human ear in low volume ranges. The INM allows users to input data on building reflections and terrain characteristics to predict noise levels around airports more accurately. However, the area around TSN airport is an area with flat terrain and mainly low-rise buildings; therefore, we did not input building and terrain data into the model calculation. The INM allows users to calculate noise levels over different time periods, such as hourly or daily intervals, depending on the needs of the analysis. However, it is important to note that the model is designed to estimate noise levels specifically generated by aircraft and does not evaluate other sources of noise exposure.

In the 2nd and 3rd surveys, the noise contour map was calculated by updating the noise map of the 1st survey and referring to the TSN airport route information on Flightradar24’s flight-tracking service. The global positioning satellite information on the respondents’ houses was used to define their locations on the map. Exact noise values were estimated for each house. The INM model was used specifically to estimate the noise level for aircraft noise only. Other sources of noise exposure were not evaluated in this study.

### 2.4. Model Development

In this study, the structural equation modeling (SEM) was used to analyze the complex relationships between noise exposure, personal characteristics, contextual factors, and health outcomes. The SEM technique involves multiple regression analyses of factors among a single measured dependent variable and a group of predictors [28]. In SEM, confirmatory factor analysis (CFA) establishes a model fit to the data to evaluate whether unobserved variables are measured by the observed constructs, where each unobserved variable is assumed to be related to a set of observed variables. Table 2 lists the measured variables and the evaluation scales used in the SEM model.

In this study, we investigated the effects of aircraft noise decrease in association with non-acoustic factors. The correlations between the non-acoustic and acoustic factors was explored by fitting them into the model. Annoyance is a specific combination of emotional, attitudinal, cognitive, and behavioral responses to environmental noise. Furthermore, environmental noise at night affects sleep, with immediate physiological consequences, and self-reported sleep quality. The aim was to achieve a common model for comparing the community response to noise before and after the change, and to clarify the benefit of the noise decrease perceived by Ho Chi Minh City residents. The SEM analysis was performed using SPSS Amos Version 26 software (IBM, New York, NY, USA).

The developed models were evaluated using the goodness-of-fit test, which determines model rejection or acceptance. In this study, the four most commonly reported goodness-of-fit tests were used: the chi-square, goodness-of-fit index (GFI), comparative fit index (CFI), and root mean square error of approximation (RMSEA). The GFI, which analyzes the percentage in the model co-variances, should be equal to 0.9 or higher for a parsimonious model. A CFI value close to 1 indicates an excellent model and must be ≥0.90 for a model to be accepted. RMSEA is the difference per degree of freedom, with a value of ≤0.8 indicating a good model fit.

## 3. Results

### 3.1. Demographic Data of the Survey Respondents

A total of 502, 145, and 519 responses were obtained from the 1st, 2nd, and 3rd surveys, respectively (Table 3). In all three surveys, the proportion of women was slightly higher than that of men. The proportions of respondents aged < 60 years in the three surveys reflected the characteristics of the young population of Vietnam. The second survey was conducted when the city municipality was calling on residents to implement social distancing to limit the spread of the infection. Face-to-face interviews with the interviewers were refused by many residents who had agreed to be interviewed during the first survey. As a result, only 145 of the participants in the first survey continued to respond in the second survey.

### 3.2. Noise Exposure and Community Response

The total number of flights observed per day during the investigation period was consistent with the decrease in operations. The number of flights observed during the 1st survey dropped from 728 to 413, as was observed in the 2nd survey following the decision to stop international flights in March. The number of flights decreased to 299 in the 3rd survey as the travel ban was extended to domestic passengers due to the re-emergence of the pandemic in July. The average noise levels and differences estimated between the 2019 and 2020 surveys, as listed in Table 3, show that noise changed proportionally with the fluctuation in the number of flights. Sites 5 and 6, which are closest to the TSN airport arrival and departure routes, had the highest noise levels. Compared with the sound levels measured in the 2019 survey, the surveys conducted during the pandemic in 2020 showed a significant decrease in day–evening–night weighted sound pressure levels (*L*_den_) and the equivalent nighttime continuous sound pressure levels (*L*_night_). Environmental noise at night affects sleep, causes immediate physiological consequences, and affects self-reported sleep quality. The direct effect of noise on sleep may negatively affect cognitive and daytime performances. With the results of the noise reduction, it can be predicted that the problem of sleep effects or insomnia will improve.

For clarity, the data from all the respondents in each survey are reported in Table 4, Table 5 and Table 6, including the data from those who did not continue to participate. For the 1st survey, the data on the respondents who continued to participate were included to allow for a comparison of the changes in the sleep and noise annoyance between those who continued to participate and those who did not.

Table 4 lists the percentage of highly annoyed individuals (%HA), the percentage of those with insomnia (%ISM), and the number of responses in each survey. For the 1st survey, the numbers in parentheses are the data of all the participants, and the numbers outside the parentheses are the data of the participants who participated in both the 1st and 2nd surveys. Thus, some of the data remained constant between the 1st and 2nd surveys. The sudden increase in %HA found at Site 11 in the 3rd survey needs further examination. Because this result is quite different from those of the other sites, the data from Site 11 were not included in the calculations of this study. In the 2020 survey, the highly annoyed percentage (%HA) did not decrease proportionally with a decrease in noise levels. At Site 5 in the 2020 surveys, the number of respondents with insomnia significantly increased, even though the noise reduction was remarkable.

To convey the level of noise exposure and disturbance experienced by the residents in the study area, the number of residents within the noise mapping area was classified into four ranges, as shown in Table 5. With regard to to the current noise standards in Vietnam, the maximum permissible noise levels in residential areas during the day should not exceed 55 decibels (dB), and at night, the levels should not exceed 45 dB. The WHO guideline recommendations for aircraft noise are that noise should be <45 dB for day–evening–night (*L*_den_) and <40 dB for nighttime (*L*_night_). In the second survey, the number of residents experiencing noise exceeding the maximum permissible noise level of 55 dB (*L*_den_) during the day decreased, but this number increased again in the third survey. However, with regard to nighttime, the number of residents experiencing noise above a given threshold level of 45 dB (*L*_night_) was lower in the third survey than in the 2nd survey.

Table 7 and Table 8 summarize the percentage and number of highly annoyed respondents and respondents with insomnia in all three surveys at different noise exposure ranges. The *p*-value derived by the Wald test shows that *L*_den_ was significantly correlated with %HA in the 1st survey (all datasets); in the 2nd survey at the <0.01 level; and in the 3rd survey at the <0.0001 level. Higher noise levels increased the possibility of being highly annoyed during the 1st and 2nd surveys. However, the %HA decreased as the noise level increased in the 3rd survey. This significant correlation was not found in the dataset of the 1st survey which comprised 145 respondents who participated in the 2nd survey (*p* = 0.2152). *L*_night_ was not significantly associated with the percentage of respondents with insomnia (%ISM) in any survey. This finding indicates that insomnia may be influenced more by non-acoustic factors.

It is noteworthy that in the 3rd survey the %HA at the survey sites with noise levels lower than 60 dB was higher than those with higher noise levels. Specifically, these results were observed at Sites 3 and 4. These sites are located near the eastern end of the southern runway on the landing side of the airport. However, the aircraft mainly used the north runway for landing during the survey period. This factor, combined with residential areas consisting mainly of low-floor houses, can lead to a more transparent view of closed aircraft passing at low elevation angles. This may lead to low estimated noise levels and a high degree of annoyance in these areas.

### 3.3. Variables in the Model

The model was created by integrating the questionnaire items from a socio-acoustic survey. The variables included in the initial model were synthesized from the acoustic and non-acoustic factors investigated in all the surveys, as listed in Table 9. The evaluations of the 145 respondents who participated in both the 1st and the 2nd surveys differed before and after noise change. The percentage of negative evaluations of the residential area’s environment in the 2nd survey decreased for green space, street scenery, view, work convenience, healthcare convenience, and daily life service convenience; however, it increased for quietness, education, and transport convenience, compared to the previous evaluation in the 1st survey. Meanwhile, the percentages of negative evaluations of the aspects of green space, street scenery, and views from houses increased drastically in the 3rd survey compared with the 1st survey. The ratings for quietness and convenience did not change significantly. The number of respondents sensitive to cold, noise, chemicals, dust, pollen, and polluted air decreased; however, sensitivity to heat, odor, and vibration increased in the 2nd survey. In particular, the percentage of respondents who worked reduced. The rate of staying at home for more than 8 h increased in the 2nd survey, reflecting the change in working conditions after the outbreak. The reduction in the number of respondents who exercised more than four times per week demonstrates the effect of the social distancing policy.

In this study, noise sensitivity was recognized as an important moderator that could alter the effect of environmental noise exposure on health outcomes. As such, noise sensitivity was included in all of the survey questionnaires, with one of the seven items enquiring about sensitivity with the following question: “In daily life, climatic factors as well as environmental conditions affect us much, then how much are you sensitive to the following factors?” The respondents were asked to respond to each item on a five-point scale ranging from “Not at all” to “Extremely”. The percentage of respondents who were sensitive was considered as the percentage of respondents who chose “Very” or “Extremely” out of the five-point verbal scale. In addition, residential area preference and quality were investigated using the following question: “Please evaluate your living area according to the following categories: Green, Street scenery, and view from houses”. The respondents were asked to rate their living area on a scale ranging from “extremely good” to “extremely bad”. The percentage of respondents who were not satisfied with their residential areas was considered as the percentage of respondents who chose “Very” or “Extremely bad” out of the five-point verbal scale.

In the SEM, a latent variable was constructed using a group of observed variables that indicated the same aspect. For example, personal sensitivity was created from self-reported sensitivities to several environmental conditions, including noise, cold, heat, and odors. By including both observed and latent variables, the SEM effectively investigates factors that are not directly measured. First, separate models were developed for the 2019 and 2020 surveys. The modification process involved a series of steps to achieve a common model for both the 2019 and the 2020 surveys. First, separate models were developed for each survey and then the models were compared to identify the differences and similarities between them. Trial and error was then used to modify the models by adding or removing observed variables, adjusting the relationships between variables, and examining the model fit statistics. The modification process aimed to achieve a common model that adequately represented the data from both surveys. Overall, the modification process involved an iterative approach that incorporated feedback from statistical analyses and a theoretical understanding of the relationships between the variables.

Table 10 presents the variables used to construct the final model. Sensitivity, living conditions, and health were the three latent variables that were included in the final model. Three or two observable variables were used to evaluate each of the latent variables.

### 3.4. Comparison of Noise Annoyance Models between 2019 and 2020 Surveys

As the pandemic made face-to-face interviews difficult, the number of responses in the 2nd and 3rd surveys was minimal. In the SEM analysis, the data from these two surveys were combined to represent the situation after the change which occurred in 2020, henceforth referred to as the 2020 survey. The data from the 2019 survey included in the SEM analysis consist of the responses from all the participants, not just those who continued to participate in the second survey. As shown in Figure 2, the model included three latent variables: sensitivity, health, and living conditions. Each latent variable was evaluated using two or three observable variables. Living conditions were determined based on green spaces and views from living areas. Personal sensitivity was measured based on the sensitivity to noise, vibration, and cold. Stress, sleep disturbances, and nutritional concerns were measured as health variables. The sample sizes for noise annoyance in the 2019 and 2020 surveys were 332 and 308, respectively, after removing all the responses without corresponding data from the dataset.

The variables in the model were correlated using the following relationships.

Noise annoyance was directly influenced by opening bedroom windows during the dry season, sensitivity, health, and living conditions.Living conditions were influenced directly by noise exposure (*L*_den_).Living conditions directly and indirectly influenced noise annoyance through sensitivity.Noise exposure (*L*_den_) indirectly influenced noise annoyance via the opening of bedroom windows during the dry season.Health status was directly influenced by noise exposure (*L*_den_), living conditions, and noise sensitivity.

It is worth noting that in this model, the significant path was direct from annoyance to health. Previous studies have suggested that noise exposure can result in annoyance and sleep disturbance and that these symptoms may accumulate over time and contribute to a range of mental and physical health outcomes [30,31]. However, it is important to note that the specific mechanisms by which noise exposure affects health are complex and may involve a range of biological, psychological, and social factors. During the model development process, we found that the initial model, which linked noise to annoyance, did not fit the data well. However, our revised model, which showed that noise directly affected health and subsequently influenced annoyance, achieved a better fit. This suggests that the health factors investigated in this study were defined by attitudes toward health, including awareness of nutrition, self-assessment of stress levels, and quality of sleep.

Figure 3 shows the models developed for both surveys and the analysis results. The chi-square value was statistically significant (224.970, *p* < 0.01). The GFI and CFI values were 0.941 and 0.856, respectively, for noise annoyance. The RMSEA value for the models was 0.057. The standardized regression weight annotated for each path in the models indicates the relative importance of each path and the effect size of the determinant variable on the variable in the path direction.

Noise exposure in the 2019 model indirectly affected noise annoyance through living conditions, whereas that in the 2020 model indirectly affected noise annoyance via the opening the bedroom window during the dry season and health variables. Living conditions and sensitivity directly influenced noise annoyance in the 2019 model. However, this was not observed in the 2020 model. However, the paths from the opening windows and health variables to noise annoyance were insignificant in the 2019 model but significant in the 2020 model. Table 11 presents the parameter estimates of these relationships.

The significant difference in the associated paths among the factors in the two models indicates two different causal structures of noise and community response represented by noise annoyance. Before the pandemic outbreak, the aspects of living conditions were the variables mediating the relationship between noise and annoyance in the 2019 survey. In the 2020 survey, health and frequency of opening windows became the main mediating factors for this relationship. These differences could be interpreted as an evaluation of noise annoyance by residents in Ho Chi Minh City; this annoyance was influenced by the quality of the outside living environment. After the epidemic outbreak, it depended more on self-assessed health and the indoor living environment, as measured by the frequency of the opening of bedroom windows.

Under the coronavirus-related restrictions applied in March 2020, many people in Ho Chi Minh City were requested to work at home, resulting in significant changes in their lifestyle. The observed variable of opening the bedroom window during the dry season was replaced by the length of time at home in the second model to verify whether the community response to aircraft noise in Ho Chi Minh City was affected by this change (Figure 4). After excluding answers with blank data for the variables used in the model, the sample sizes for noise annoyance in the 2019 and 2020 surveys were 332 and 308, respectively.

The chi-squared value was statistically significant (chi-square = 269.964, *p* < 0.01). The GFI and CFI were 0.935 and 0.826, respectively, for noise annoyance. The RMSEA value for the model was 0.064. The noise annoyance of the 2020 model was directly affected by the length of time at home and indirectly by the noise level (*L*_den_). Noise annoyance was directly affected by health in the 2020 survey, but not in the 2019 model. In a similar manner to the frequency of window opening, during the epidemic a longer time at home could increase noise annoyance. Negative personal health assessments also increased the effect of noise. Table 12 presents the parameter estimates of these relationships. The results of the 2019 model are consistent with the findings of a previous study using data from a survey conducted around TSN in 2008, which determined that satisfaction with the living environment measured by the preference for living areas was the main modifier of noise annoyance [32].

### 3.5. Comparison of Insomnia Model between the 2019 and 2020 Surveys

As shown in Figure 5, the final structural model developed for insomnia in the 2019 and 2020 surveys included three latent variables: sensitivity, health, and living conditions. Each latent variable was evaluated using three observable variables. Sensitivity to noise, vibration, and odors determined personal sensitivity. Health was assessed based on stress, sleep disturbances, and nutrition. The views from living spaces, green spaces for living areas, and street scenery influenced the evaluation of living conditions. After excluding answers with blank data, the sample sizes for insomnia in the 2019 and 2020 surveys were 295 and 291, respectively.

The variables in the model were correlated with the following relationships:Insomnia was influenced by sensitivity, health, living conditions, and the opening of the bedroom window during the dry season.Noise exposure (*L*_night_) had a direct impact on living conditions.Insomnia was influenced directly or indirectly by living conditions through sensitivity.Noise exposure (*L*_night_) indirectly influenced insomnia via the opening of bedroom windows during the dry season.Noise exposure (*L*_night_), living conditions, and sensitivity had an impact on health.

Numbered lists can be added as follows:

The chi-squared value was statistically significant (chi-square = 275.578, *p* < 0.01). The GFI and CFI for insomnia were 0.927 and 0.894, respectively. The RMSEA value for the models was 0.058.

Figure 5 shows that insomnia in the 2019 model was indirectly affected by noise exposure (*L*_night_) through the observed variable of the opening of windows in the dry season. In the 2020 model, insomnia was indirectly affected by night noise exposure (*L*_night_). Sensitivity had no impact on insomnia in the 2019 model but directly and indirectly affected insomnia in the 2020 model. Table 13 summarizes the parameter estimates for these relationships.

Similarly, we constructed a new model by changing the observed variable from the opening of the bedroom window during the dry season to the length of time at home, as shown in Figure 6. The sample sizes for insomnia in the 2019 and 2020 surveys were 295 and 291, respectively, after removing all responses without corresponding data from the dataset. The chi-squared value was statistically significant (chi-square = 279.004, *p* < 0.01). The GFI and CFI were 0.928 and 0.892, respectively, for insomnia. The RMSEA value for the models was 0.058.

In the 2019 model, the observed and latent variables had no direct or indirect influence on insomnia. In contrast, in the 2020 model, insomnia was indirectly affected by nighttime noise exposure (*L*_night_) through health. Insomnia was influenced by sensitivity both directly and indirectly. All the above-modified relationships are summarized in Table 14. It should be noted that the sensitivity exhibited a strong regression with insomnia and health in all the 2020 models. This regression was not significant in the 2019 model. The sensitivity was found to be significantly influenced by living conditions in all the models of the 2019 survey. This linking path showed no importance in any of the models of the 2020 survey. The path linking noise exposure to insomnia through health was essential after the epidemic outbreak but not in 2019. In other words, the sleep effects of nighttime noise were more evident in 2020 than in 2019. Unlike annoyance, insomnia cannot be resolved by an improved assessment of the living environment but by personal factors such as health and sensitivity.

## 4. Discussion

### 4.1. Causal Structure of Noise Annoyance and Insomnia in the Context of the Noise Reduction

The context of the decreased operation at TSN during the coronavirus crisis provided an opportunity to assess the effects of a significant change in aircraft noise exposure on community reactions and public health in Ho Chi Minh City. The structural equation model (SEM) proposed in this study presents the correlation between non-acoustic and acoustic factors and clarifies how these relations define the response to aircraft noise in residential areas around the airport. A common model was developed for the 2019 and 2020 surveys to compare the structures of community responses to noise situations before and after the change. According to the parameter estimates of the models, noise exposure had indirect effects on noise annoyance and insomnia via non-acoustic factors. This finding indicates that, like noise exposure, non-acoustic variables significantly influence noise annoyance and insomnia. This finding supports previous studies that confirmed that aircraft noise can have adverse effects on health, including annoyance, sleep disturbance, cardiovascular disease, and cognitive impairment [33]. Noise pollution can have both auditory and non-auditory effects on health, including cardiovascular disease, sleep disturbance, and annoyance [31,34]. There was a significant association between self-reported sleep disturbance and environmental noise, with a stronger effect observed for transportation noise than for neighborhood noise [35]. Janssen et al. (2011) also suggested that annoyance is influenced by factors beyond noise level [36]. Environmental noise can have negative effects on sleep quality and duration, with transportation noise being the most prevalent source of a decreased quality of life [37].

Noise sensitivity is an important factor shaping residents’ perceptions of the acoustic environment. Living conditions and sensitivity were influential variables in the annoyance model for 2019, whereas health and sensitivity were significantly linked to insomnia in the 2020 model. This result is in agreement with a previous study in which noise sensitivity was associated with annoyance and a lower health-related quality of life in adults exposed to environmental noise [38]. Similarly, Croy et al. revealed that people’s subjective experience of noise was reflective of their physiological responses [39].

The length of time spent at home variable was added to the second model to verify whether the response of the residents of Ho Chi Minh City to noise was affected by lifestyle changes due to the pandemic. It is worth noting that the paths linking the length of time spent at home, starting from noise and moving toward annoyance, are significant in the 2020 model, indicating that spending a longer time at home increases the community’s negative response to noise. Staying at home for longer periods may have made residents more sensitive to their living environment and surroundings.

Furthermore, in the 2020 survey, health was found to significantly affect noise annoyance and insomnia. Before the epidemic outbreak, noise affected sleep through the frequency of window opening; however, after that, noise affected health and sleep in 2020. The health variables in this study’s model were sleep trouble, nutritional interest, and stress levels. The finding that health can increase annoyance and insomnia can explain why the percentage of highly annoyed individuals and insomnia did not decrease with the corresponding significant decrease in aircraft noise. A study on the impact of the COVID-19 lockdown measures on noise levels in urban areas in the Ruhr area of Germany indicates that the lockdown measures resulted in a significant reduction in noise levels, particularly in the frequency range of human speech, and that this reduction was most noticeable during the daytime and in residential areas [40]. This change may have made aircraft noise a more noticeable noise source in urban residential areas despite its remarkable decrease in its average level aspect.

### 4.2. Effect of COVID-19 Pandemic on Community Perception of Aircraft Noise

The results of this study suggest that during the pandemic, people’s increased concerns about the global situation may have overshadowed the stress caused by noise exposure, which could explain why the insomnia data remained unchanged. A report by the European Environment Agency suggested that during the COVID-19 pandemic, people’s attention may have shifted away from environmental noise and toward other stressors related to the pandemic, such as health concerns, economic uncertainty, and social isolation [41]. Several studies on people’s perception of environmental noise during the pandemic have found that people who were more concerned about the pandemic reported lower levels of annoyance and more positive attitudes toward noise compared to those who were less concerned [42].

On the other hand, this study demonstrated the mental health burden of the global pandemic regarding aviation noise. These findings differed from those of recent studies on the mental health burden of the global pandemic of the coronavirus in Europe relating to the decrease in traffic noise exposure due to the coronavirus pandemic. Wojciechowska et al. found that long-term exposure to aircraft noise was associated with higher blood pressure and arterial stiffness, and that the COVID-19 lockdown period, which resulted in reduced aircraft noise exposure, was associated with lower blood pressure and arterial stiffness [43].

### 4.3. Study Strengths and Limitations

The strength of this research lies in its investigation of the impact of COVID-19 on the community response to noise before and after flight operation was restricted; it provides valuable data for understanding the noise exposure decease intervention and its effect on health. A large sample size of approximately 1200 responses from 12 residential areas around Tan Son Nhat Airport increased the reliability of the findings. Structural equation models (SEMs) were employed to investigate the relationship between noise annoyance, insomnia, and health, providing a comprehensive analysis of the factors affecting community response to noise. The study used a common model of noise annoyance and insomnia, which corresponded to the situation before and after the change, enabling the comparison of the results.

However, this research has some limitations. It only investigated the impact of the COVID-19 event, limiting the generalizability of the findings to other situations. Additionally, it focuses only on one airport, limiting the applicability of the findings to other airports and regions. The data collection relied on a survey that may have been affected by response bias and did not capture objective measures of noise exposure. In our study, the prediction of noise exposure was performed using an INM. The prediction using an INM does not consider low-frequency sounds below 50 Hz. Low-frequency sounds, also known as infrasound, are often associated with industrial and transportation activities, such as aircraft, wind turbines, trains, and heavy machinery. While infrasound is not audible to humans, it can cause health problems, including nausea, dizziness, and headaches. Several studies have revealed the impact of infrasound on sleep and the potential sources of formation, including aircraft noise during takeoff and landing. A review by Basner et al. (2014) found that aircraft noise, including the low-frequency components of noise, can cause sleep disturbances and insomnia in people living near airports, particularly during takeoff and landing [44]. Exposure to environmental noise, including low-frequency noise, can cause sleep disturbances and insomnia, as well as other health effects, such as cardiovascular disease, hypertension, and cognitive impairment. However, Omlin et al. (2019) found that infrasound emitted by wind turbines did not have a significant impact on sleep quality or quantity in the general population [45]. There is some evidence linking low-frequency noise and infrasound to a range of health effects, including sleep disturbances [46]. Further research is needed to confirm these associations with the aircraft noise impact and to better understand the underlying mechanisms.

Therefore, the limitation of the INM when considering low-frequency sounds below 50 Hz may underestimate the actual noise exposure levels in certain situations. It is essential to consider this limitation when using the INM and to supplement its predictions with other measurements and assessments, especially in situations where low-frequency sounds are prevalent. Aircraft noise emits low-frequency sounds with directional characteristics in the lateral and rearward directions during takeoffs and landings, respectively [47,48,49]. However, because data on the exposure and impact of low-frequency sounds below 50 Hz for the aircraft models currently operating at Tan Son Nhat airport are not available, it is impossible to evaluate the actual exposure and impact of infrasound. We intend to address this issue in future research.

We acknowledge that the decision to remove data from Site 11 could be related to the pandemic’s influence and that the potential bias caused by this decision was not fully examined in the current study. Site 11 may have experienced unique circumstances that led to the sudden increase in %HA, which may not be representative of other sites or populations. Therefore, caution should be exercised when interpreting the results of our study, particularly with the exclusion of the data from Site 11. Furthermore, we acknowledge that the pandemic could affect noise annoyance and sleep for the survey population in general, which is an important limitation of the study. While we attempted to control for the pandemic’s impact using data from similar periods in 2019 and 2020, there may still be confounding factors that were not accounted for. Future research should investigate the impact of the pandemic on noise annoyance and sleep in greater detail to better understand the potential limitations.

Lastly, the study did not consider other factors that may affect a community’s response to noise, such as social and economic factors; this limits the comprehensiveness of the analysis.

## 5. Conclusions

This study assessed the impact of COVID-19 on the relationship between noise annoyance and insomnia and investigated the causal structure of noise annoyance and insomnia in the context of noise decrease. This study used a structural equation model (SEM) to analyze the correlation between non-acoustic and acoustic factors and how they define the response to aircraft noise in the residential areas around Tan Son Nhat Airport in Ho Chi Minh City. The findings indicated that noise exposure had indirect effects on noise annoyance and insomnia via non-acoustic factors and that non-acoustic variables significantly influenced noise annoyance and insomnia as much as noise exposure. This study revealed a significant and direct path from annoyance to health in the model. This finding is particularly noteworthy given the accumulating evidence linking noise exposure to annoyance and the subsequent negative mental and physical health outcomes. Our results also suggest that the specific mechanisms through which noise exposure affects health are complex and multifaceted and involve biological, psychological, and social factors. Overall, this study provides important insights into the relationship between noise exposure, annoyance, and health outcomes. Our findings highlight the need for continued research into the underlying mechanisms of these relationships as well as the importance of promoting strategies to mitigate noise exposure and improve health outcomes.

The study also revealed that noise sensitivity, health, and lifestyle change due to the pandemic were influential variables in the annoyance and insomnia models. Additionally, the study showed that health can increase annoyance and insomnia and explain why the percentage of highly annoyed individuals and insomnia did not decrease despite a significant decrease in aircraft noise during the pandemic. However, people’s concerns about the global situation during the pandemic may have replaced the stress related to noise exposure, causing the insomnia data to remain unchanged.

This study provides valuable insights into the effects of decreased aircraft noise exposure on the residents’ self-reported health status near TSN and supplements the limited data available on the health impacts of noise intervention in developing countries. Our findings have important implications for policymakers seeking to manage aircraft noise in a way that is protective of residents’ health and quality of life. The results of this study underscore the urgent need for continued research on the health impacts of environmental noise, including aircraft noise, and the development of evidence-based policies and interventions to mitigate these impacts. Our findings also highlight the importance of engaging with local communities to understand their experiences and priorities when developing noise management strategies.

Overall, our study contributes to the growing body of research on the health impacts of environmental noise and demonstrates the feasibility and effectiveness of noise intervention programs in improving residents’ health outcomes. Our findings have broad applicability across different cultures and economies and can contribute to the development of the WHO guidelines for noise management. In conclusion, our study provides critical insights into the relationship between aircraft noise exposure and residents’ health, highlighting the importance of proactive and evidence-based noise management strategies that prioritize the health and well-being of local communities. We hope that our findings serve as a catalyst for further research and policy development in this important area.

## Figures and Tables

**Figure 1 ijerph-20-05450-f001:**
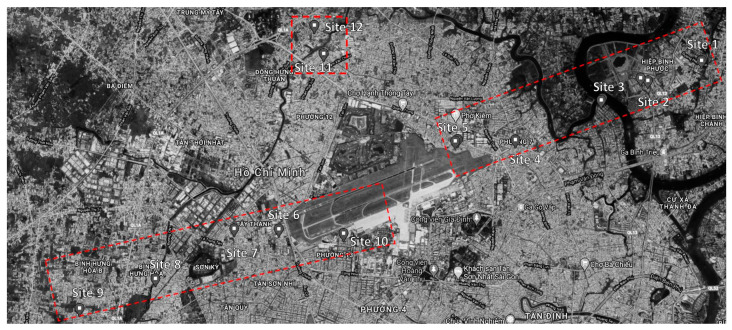
Map of survey sites indicating the location of the surveyed residential areas.

**Figure 2 ijerph-20-05450-f002:**
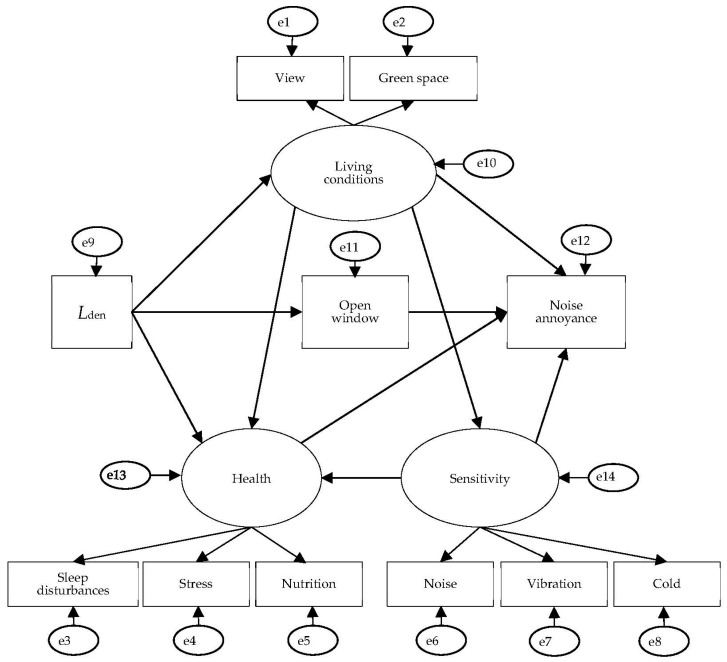
The final version of the structural equation model estimated for noise annoyance in the 2019 and 2020 surveys.

**Figure 3 ijerph-20-05450-f003:**
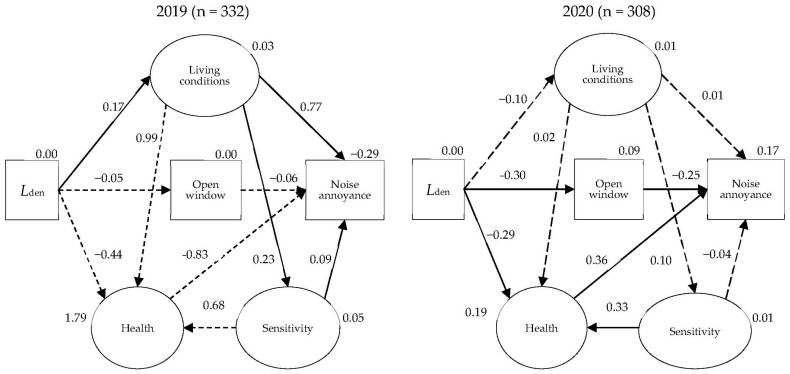
The structural equation model estimated the noise annoyance with the opening of the bedroom window variable in the 2019 and 2020 surveys using chi-square, GFI, CFI, and RMSEA statistics: chi-square = 224.970, *p* < 0.01, df = 74, GFI = 0.941, CFI = 0.856, and RMSEA = 0.057. Statistically significant paths and standardized regression weights are annotated with (*p* < 0.05). The non-significant paths are represented by dashed lines. The explained variances are annotated for each variable.

**Figure 4 ijerph-20-05450-f004:**
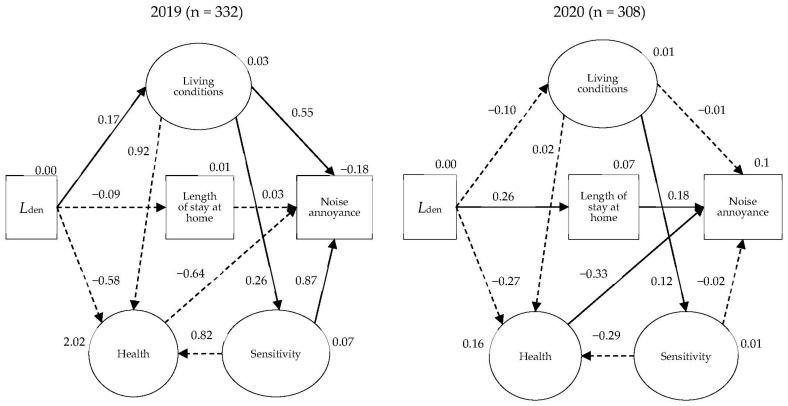
Structural equation model estimated for noise annoyance with length of time at home variable in the 2019 and 2020 surveys using chi-square, GFI, CFI, and RMSEA statistics: chi-square = 269.964, *p* < 0.01, df = 74, GFI = 0.935, CFI = 0.826, and RMSEA = 0.064. Statistically significant paths and standardized regression weights were annotated with (*p* < 0.05). The non-significant paths are represented by dashed lines. The explained variances are annotated for each variable.

**Figure 5 ijerph-20-05450-f005:**
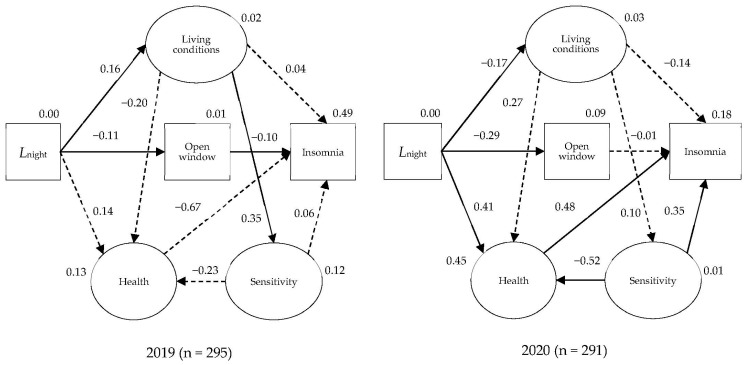
The structural equation model estimated for insomnia with the opening of the bedroom window variable in the 2019 and 2020 surveys using chi-square, GFI, CFI, and RMSEA statistics: chi-square = 275.578, *p* < 0.01, df = 94, GFI = 0.927, CFI = 0.894, and RMSEA = 0.058. Statistically significant paths and standardized regression weights were annotated with (*p* < 0.05). The non-significant paths are represented by dashed lines. The explained variances are annotated for each variable.

**Figure 6 ijerph-20-05450-f006:**
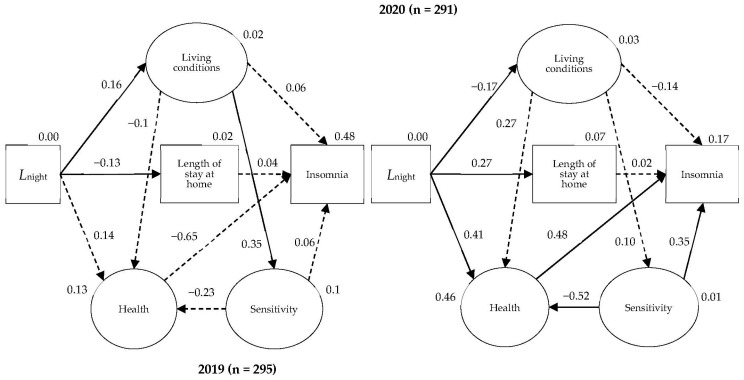
Structural equation model estimated for insomnia with length of time at home variable in the 2019 and 2020 surveys using chi-square, GFI, CFI, and RMSEA statistics: chi-square = 279.004, *p* < 0.01, df = 94, GFI = 0.928, CFI = 0.892, and RMSEA = 0.058. Statistically significant paths and standardized regression weights were annotated with (*p* < 0.05). The non-significant paths are represented by dashed lines. The explained variances are annotated for each variable.

**Table 1 ijerph-20-05450-t001:** Questions and measuring scales used to identify annoyance and insomnia among respondents.

Annoyance
Question: Thinking about the last 12 months (1st survey)/3 months (2nd survey)/4 months (3rd survey) or so, what number from 0 to 10 best shows how much you are bothered, disturbed, or annoyed by aircraft noise?
Evaluation scale: 11-point numerical scale from 0 (not annoyed at all) to 10 (extremely annoyed)
Insomnia
Question: Please answer this question regarding your sleep:
(a) Do you have any trouble sleeping? No/Yes
(b) If you answered “Yes” to the above question, please choose the corresponding alternative (alternatives: rarely or not at all; once or twice a week; more than three times a week) for the following items: (1) it is difficult to fall asleep; (2) when awakened during the night, it is difficult to sleep again; (3) awakened early in the morning; (4) do not wake the next morning with the feeling of having slept well; (5) sleepy during daytime and cannot work well; (6) other
Evaluation scale: 1: have no insomnia symptoms (*); 2: have insomnia symptoms
(*) Respondents with insomnia symptoms responded affirmatively to question (a): Do you have any trouble with your sleep? They also responded with (5), sleepiness during daytime and inability to work well more than three times a week, and that they had experienced at least one of the other symptoms (1)–(4) more than three times per week.

**Table 2 ijerph-20-05450-t002:** Latent and observed variables and their evaluation scale used in the initial SEM model.

Latent Variable	Observed Variable	Question	Scale
Residential factors	Housing type	What type of house ownership do you have?	1: Owned 2: Rented 3: Other
	Housing structure	Structure of the house?	1: Wooden 2: Bricks 3: Prefabricated 4: Reinforced concrete 5: Reinforced concrete with brick wall 6: Other
	Number of glass layers in living room windows and doors	How many glass layers do your living room windows and doors have? If they are double-paned windows/doors, please specify as 2 layers.	1: More than 3 layers 2: 2 layers 3: 1 layer 4: Other
	Type of frame of living room windows and doors	Which type of frame among the following types do your living room windows and doors have?	1: Aluminum frame 2: Wooden frame 3: Plastic frame 4: Other
	Number of glass layers in bedroom windows and doors	How many glass layers do your bedroom windows and doors have? If they are multi-layer double-paned windows/doors, please specify as 2 layers.	1: More than 3 layers 2: 2 layers 3: 1 layer 4: Other
	Type of frame of bedroom windows and doors	Which type of frame among the following types do your bedroom windows and doors have?	1: Aluminum frame 2: Wooden frame 3: Plastic frame 4: Other
Coping capacity	Opening of bedroom windows (% Often and Always)	Do you open bedroom windows while sleeping in the dry/rainy season?	1: Rarely to 4: Always
	Number of hours staying at home	Thinking about the last 4 months, how much of the day do you spend at home?	1: Under 8 h 2: 8–15 h 3: Over 15 h hours
Health	Effects on sleep	How often do you have any trouble getting to sleep or staying asleep?	1: Often 2: Sometimes 3: Almost never
	Stress	Thinking about the amount of stress in your life, how stressful would you say that most days are?	0: Not at all to 10: Extremely
	Life satisfaction	How do you feel about your life as a whole right now?	0: Very dissatisfied to 10: Satisfied
	Self-reported health status	Do you usually have periodic health examination?	1: No 2: Yes
	Awareness of nutrition balance	Do you think about the nutritional balance of your diet?	1: Do not think about it to 4: Think a lot
	Salt intake	Do you eat or drink salty food or soup?	1: Not often 2: Once a day 3: Every meal
	Alcohol intake	How often do you drink alcohol?	1: Not at all 2: 2–3 times a month or less 3: 1–4 days a week 4: Almost every day
	Smoking habit	Do you smoke cigarettes?	1: Not at all 2: I smoked before but stopped 3: 20 or less per day 4: More than 20 per day
	Exercise frequency	How often do you engage in physical activity over 30 min?	1: Almost everyday 2: 4–5 times a week 3: 2–3 times a week 4: About once a week 5: Once or twice a month 6: Not at all
	Morbidity	Have you ever had any of these conditions? 1. Heart trouble 2. High blood pressure or hypertension 3. Hyperlipidemia 4. Stroke, small stroke, or TIA 5. Asthma 6. Diabetes 7. Cancer 8. Depression or neurosis 9. Other	0: No diseases 1: Diseases
	Body Mass Index (BMI)	What is your height and weight?	0: BMI < 29 1: BMI > 29 (obese)
Personal and attitudinal factors		In daily life, how sensitive are you to the following climatic factors and environmental conditions	
	Noise	Noise?	1: Not at all to 5: Extremely
	Cold	Cold?	1: Not at all to 5: Extremely
	Heat	Heat?	1: Not at all to 5: Extremely
	Odors	Odors?	1: Not at all to 5: Extremely
	Vibration	Vibration?	1: Not at all to 5: Extremely
	Sex	Gender of respondent?	1: Male 2: Female
	Job	What is your job?	1. Employed 2. Student 3. Homemaker 4. Retired 5. Unemployed
	Age	How old are you?	0: <60 years old 1: >=60 years old
	Residence period length	How long have you been living in your present house?	0: >5 years 1: <=5 years
Living conditions		Please evaluate your living area according to the following items:	
	Green space	Green space?	1: Extremely good to 5: Extremely bad
	Street scenery	Street scenery?	1: Extremely good to 5: Extremely bad
	View	View?	1: Extremely good to 5: Extremely bad
	Quietness	Quietness?	1: Extremely good to 5: Extremely bad
	Work convenience	Work convenience?	1: Extremely good to 5: Extremely bad
	Education convenience	Education convenience?	1: Extremely good to 5: Extremely bad
	Health care convenience	Health care convenience?	1: Extremely good to 5: Extremely bad
	Daily life service convenience	Daily life service convenience?	1: Extremely good to 5: Extremely bad
	Transport convenience	Transport convenience?	1: Extremely good to 5: Extremely bad

**Table 3 ijerph-20-05450-t003:** Demographic characteristics of the survey respondents, including age, gender, length of residence, and occupation.

		1st Survey	2nd Survey	3rd Survey	Vietnam Census (2019) *
Number of respondents		502	145	519	
Response rate (%)		60.3	28.9	68.6	
Sex	Male	46.2	46.5	49.2	49.9
	Female	53.8	53.5	50.8	50.1
Age	<60 years old	81.9	70.6	89.9	88.1
	≥60 years old	18.1	29.4	10.1	11.9
Length of Residence	0–5 years	51.1	27.7	40.0	
	More than 5 years	48.9	72.3	60.0	
Occupation	Employed	53.6	37.4	40.0	55.5
	Student, housewife, retired, unemployed	46.4	62.6	60.0	44.5

(*): Adapted with permission from ref. [29]; 2019 copyright by General Statistics Office of Vietnam.

**Table 4 ijerph-20-05450-t004:** Average noise levels (dB) and standard deviation (SD) estimated for each survey site during the survey period.

Site	*L*_den_ ^a^ (in Parentheses: SD)			*L*_night_ ^b^ (in Parentheses: SD)			∆*L* _den_		∆*L*_night_	
	1st (SD)	2nd (SD)	3rd (SD)	1st (SD)	2nd (SD)	3rd (SD)	2nd–1st	3rd–1st	2nd–1st	3rd–1st
1	66 (0.9)	61 (1.1)	60 (0.9)	58 (0.9)	52 (1.1)	52 (0.9)	−5	−6	−6	−6
2	64 (0.9)	61 (1.1)	61 (0.9)	57 (0.9)	52 (1.1)	53 (0.9)	−3	−3	−5	−3
3	64 (2.3)	60 (3.0)	59 (1.6)	56 (2.3)	51 (3.0)	51 (1.5)	−4	−5	−5	−5
4	62 (1.1)	57 (1.2)	57 (1.1)	55 (1.1)	48 (1.2)	49 (1.1)	−5	−6	−6	−6
5	81 (1.4)	76 (1.2)	73 (1.8)	73 (1.8)	67 (1.3)	66 (1.7)	−5	−7	−7	−8
6	75 (0.4)	71 (0.5)	69 (0.7)	67 (1.2)	61 (0.5)	61 (0.7)	−4	−6	−6	−6
7	69 (0.6)	65 (0.7)	64 (1.5)	61 (0.6)	56 (0.6)	56 (1.4)	−4	−5	−5	−5
8	66 (0.2)	62 (0.1)	62 (0.1)	58 (0.2)	53 (0.1)	54 (0.1)	−4	−4	−6	−5
9	64 (0.2)	59 (0.3)	60 (0.2)	57 (1.4)	50 (0.3)	52 (0.2)	−5	−4	−7	−5
10	67 (1.3)	62 (1.8)	65 (0.6)	59 (1.5)	54 (1.7)	57 (0.6)	−5	−2	−6	−2
11	47 (0.5)	43 (0.2)	43 (0.3)	40 (0.5)	34 (0.2)	36 (0.3)	−5	−4	−6	−4
12	45 (0.9)	41 (0.2)	41 (0.2)	38 (0.9)	33 (0.2)	34 (0.2)	−4	−4	−5	−4

^a^ Day–evening–night weighted sound pressure level; ^b^ nighttime equivalent continuous sound pressure level.

**Table 5 ijerph-20-05450-t005:** Comparison of number of responses across different noise level ranges in the three surveys.

		Noise Level Ranges *L*_den_ ^a^ (dB)			
		0–44	45–50	51–55	56–81
1st survey (all data)	%	0.0	12.2	0.0	87.8
	Response number/N	0/502	61/502	0/502	441/502
1st survey (145 data)	%	0.0	11.7	0.0	88.3
	Response number/N	0/502	17/145	0/145	128/145
2nd survey	%	12.4	0.0	29.7	57.9
	Response number/N	18/145	0/145	43/145	84/145
3rd survey	%	17.0	0.0	0.0	83.0
	Response number/N	88/519	0/519	0/519	431/519
		Noise Level Ranges *L*_night_ ^b^ (dB)			
		0–39	40–45	46–51	52–79
1st survey (all data)	%	5.8	6.4	0.0	87.8
	Response number/N	29/502	32/502	0/502	441/502
1st survey (145 data)	%	1.4	10.3	0	88.3
	Response number/N	2/145	15/145	0/145	128/145
2nd survey	%	11.7	0.0	31.1	57.2
	Response number/N	17/145	0/145	45/145	83/145
3rd survey	%	17.0	0.0	27.7	55.3
	Response number/N	88/519	0/519	144/519	287/519

^a^ Day–evening–night weighted sound pressure level; ^b^ nighttime equivalent continuous sound pressure level.

**Table 6 ijerph-20-05450-t006:** Percentage of highly annoyed (%HA) and percentage of insomnia (%ISM) among respondents and the total number of responses collected at each survey site.

Site	1st Survey (in Parentheses: All Data)			2nd Survey			3rd Survey		
	%HA ^a^	%ISM ^b^	N ^c^	%HA ^a^	%ISM ^b^	N ^c^	%HA ^a^	%ISM ^b^	N ^c^
1	0.0 (0.0)	0.0 (0.0)	5 (49)	40.0	0.0	5	2.0	11.9	50
2	0.0 (7.3)	0.0 (2.6)	2 (44)	0.0	0.0	2	17.1	0.0	35
3	0.0 (0.0)	40.0 (6.5)	5 (31)	0.0	20.0	5	28.6	2.0	49
4	0.0 (2.0)	8.3 (2.0)	36 (50)	2.8	11.1	36	9.1	0.0	44
5	0.0 (3.0)	0.0 (0.0)	10 (33)	30.0	10.0	10	7.9	5.3	38
6	7.7 (18.4)	7.7 (6.4)	13 (50)	7.7	7.7	13	2.4	0.0	42
7	13.0 (12.5)	0.0 (0.0)	24 (50)	4.3	0.0	24	0.0	0.0	40
8	8.3 (6.3)	0.0 (2.8)	14 (36)	0.0	0.0	14	4.0	0.0	50
9	0.0 (0.0)	0.0 (2.4)	7 (48)	0.0	20.0	7	4.0	0.0	50
10	10.0 (2.2)	9.1 (2.0)	12 (50)	8.3	12.5	12	3.0	3.2	33
11	0.0 (0.0)	0.0 (3.3)	15 (32)	0.0	0.0	15	72.5	0.0	40
12	0.0 (0.0)	0.0 (0.0)	2 (29)	0.0	0.0	2	0.0	0.0	48
Total	4.4 (4.8)	5.1 (2.3)	145 (502)	7.6	7.5	145	12.1	1.8	519

^a^ Percentage of respondents who were highly annoyed; ^b^ percentage of respondents who had insomnia; ^c^ number of responses.

**Table 7 ijerph-20-05450-t007:** Comparison of the percentages of highly annoyed respondents (%HA) across different noise level ranges.

		Noise Level Ranges *L*_den_ ^a^ (dB)				*p*-Value
		0–59	60–64	65–69	70–81	
1st survey (all data)	%HA	0.0	0.7	4.0	12.0	0.0012 *
	Response number/N	0/61	1/147	6/161	16/133	
1st survey (145 data)	%HA	0.0	0.0	5.4	8.7	0.2152 (n.s)
	Response number/N	0/17	0/45	2/37	4/46	
2nd survey	%HA	1.6	7.9	4.3	17.4	0.0085 *
	Response number/N	1/61	3/38	1/23	4/23	
3rd survey	%HA	15.5	13.8	0.0	5.0	<0.0001 *
	Response number/N	36/232	23/167	0/40	4/80	

^a^ Day–evening–night weighted sound pressure level. * *p*-Values < 0.01.

**Table 8 ijerph-20-05450-t008:** Comparison of percentage of insomnia respondents (%ISM) across different noise level ranges.

		Noise Level Ranges *L*_night_ ^a^ (dB)				*p*-Value
		0–54	55–59	60–64	65–79	
1st survey (all data)	%ISM	1.6	2.3	2.0	3.6	0.5049 (n.s)
	Response number/N	1/61	6/258	2/100	3/83	
1st survey (145 data)	%ISM	0.0	7.1	0.0	0.0	0.4691 (n.s)
	Response number/N	0/17	5/70	0/35	0/23	
2nd survey	%ISM	9.8	0.0	0.0	0.0	0.5768 (n.s)
	Response number/N	9/92	0/23	0/13	0/10	
3rd survey	%ISM	1.9	1.7	0.0	5.3	0.1478
	Response number/N	7/366	1/73	0/42	2/38	

^a^ Nighttime equivalent continuous sound pressure level.

**Table 9 ijerph-20-05450-t009:** Data on non-acoustic factors investigated in the surveys, including the proportion of respondents (in parentheses: number of responses) in each category, such as smoking status, employment status, and other health-related factors.

Factors	Categories	1st Survey—All Data	1st Survey—145 Data	2nd Survey	3rd Survey
Residential factors					
Housing type	Owned	64.9 (321)	76.9 (110)	※	78.4 (407)
Floor Area/Width of house	≤50 m^2^	59.1 (269)	65.9 (89)	※	66.2 (129)
Housing structure	1. Wooden	2.1 (7)	0 (0)	※	0.8 (4)
	2. Brick	14.5 (49)	7.6 (7)		25.1 (124)
	3. Prefabricated	0.3 (1)	0 (0)		0.6 (3)
	4. Reinforced concrete	44.8 (151)	52.2 (48)		20.4 (101)
	5. Reinforced concrete with brick wall	34.4 (116)	37 (34)		53.1 (263)
	6. Other	3.9 (13)	3.3 (3)		0.0 (0)
Number of glass layers in living room windows and doors	1. More than 3 layers	2.7 (13)	1.4 (2)	※	3.3 (16)
	2. 2 layers	18.2 (89)	19.3 (27)		18.6 (91)
	3. 1 layer	75.3 (369)	74.3 (104)		71.8 (351)
	4. Other (the window has no glass)	3.9 (19)	5 (7)		6.1 (30)
Type of frame of living room windows and doors	1. Aluminum frame	31.7 (156)	24.8 (35)	※	42.2 (213)
	2. Wooden frame	14.2 (70)	7.1 (10)		16 (81)
	3. Plastic frame	1.4 (7)	0 (0)		1.4 (7)
	4. Other	52.6 (259)	68.1 (96)		40.4 (204)
Number of glass layers in bedroom windows and doors	1. More than 3 layers	1.2 (6)	37 (51)	※	57.2 (214)
	2. 2 layers	13.5 (66)	6.5 (9)		16.6 (62)
	3. 1 layer	77.7 (379)	0 (0)		1.9 (7)
	4. Other (the window has no glass)	7.6 (37)	56.5 (78)		24.3 (91)
Type of frame of bedroom windows and doors	1. Aluminum frame	37.3 (181)	0 (0)	※	2.2 (8)
	2. Wooden frame	20.0 (97)	12.9 (18)		25.1 (91)
	3. Plastic frame	2.7 (13)	77.7 (108)		64.2 (233)
	4. Other	40.0 (194)	9.4 (13)		8.5 (31)
Personal and attitudinal factors					
Sex	Male	46.2 (229)	46.5 (66)	※	49.2 (255)
Age	≥60 years old	18.1 (90)	25.4 (36)	※	10.1 (52)
Residence length	≤5 years	41.7 (204)	27.7 (39)	※	40.0 (204)
Residential area preference and quality (% bad and extremely bad)	1. Green space	12.3 (60)	12.9 (18)	4.9 (7)	21.4 (110)
	2. Street scenery	7.9 (38)	7.2 (10)	3.5 (5)	16.7 (83)
	3. View from houses	8.0 (39)	7.9 (11)	7.0 (10)	16.1 (80)
	4. Quietness	9.0 (43)	13.0 (18)	21.1 (30)	9.0 (45)
	5. Work convenience	3.8 (18)	2.2 (3)	1.4 (2)	2.0 (10)
	6. Education convenience	1.9 (9)	0.7 (1)	2.8 (4)	2.0 (10)
	7. Health care convenience	3.4 (16)	2.9 (4)	1.4 (2)	2.8 (14)
	8. Daily life service convenience	1.3 (6)	1.5 (2)	0.7 (1)	2.0 (10)
	9. Transport convenience	4.4 (21)	2.9 (4)	10.5 (15)	6.6 (33)
Opening of bedroom windows (% often and always)	1. Dry season	31.2 (140)	15.9 (20)	※	45.1 (233)
	2. Rainy season	17.9 (81)	28.0 (35)		32.5 (166)
Sensitivity (% very and extremely)	1. Cold	2.9 (14)	1.4 (2)	1.6 (2)	2.2 (11)
	2. Heat	15.6 (75)	12.1 (17)	17.5 (22)	36.7 (177)
	3. Noise	16.1 (78)	16.4 (23)	14.3 (20)	13.7 (67)
	4. Vibration	8.5 (41)	7.9 (11)	10.9 (15)	6.6 (31)
	5. Chemicals	5.4 (26)	5.0 (7)	0.8 (1)	3.4 (16)
	6. Odors	8.8 (42)	8.6 (12)	8.9 (11)	12.9 (62)
	7. Dust, pollen, polluted air	6.7 (32)	8.6 (12)	5 (8)	2.6 (12)
Job	1. Employed	53.6 (266)	51.8 (72)	37.4 (52)	40.0 (207)
	2. Student	9.3 (46)	5.0 (7)	0 (0)	4.3 (22)
	3. Homemaker	13.1 (65)	15.8 (22)	4.3 (6)	16.2 (84)
	4. Retired	9.7 (48)	15.8 (22)	15.8 (22)	6.8 (35)
	5. Unemployed	14.3 (71)	14.4 (2)	13.7 (19)	32.7 (169)
Number of hours staying at home	1. Under 8 h	30.6 (149)	21.6 (30)	7.7 (11)	14.2 (72)
	2. From 8 to 15 h	36.6 (178)	35.3 (49)	41.3 (59)	60.6 (307)
	3. Above 15 h	32.6 (159)	43.2 (60)	50.3 (72)	24.9 (126)
Life satisfaction	Very dissatisfied	1.0 (5)	0.7 (1)	4.2 (6)	0.8 (4)
Health-related factors					
Self-rated health status	Fair or Poor	23.6 (115)	25.4 (36)	25.9 (37)	9.3 (47)
Stress	Quite or extremely stressful	8.0 (39)	0.0 (0)	3.5 (5)	5.6 (28)
Morbidity	1. Heart trouble	5.5 (24)	7.6 (9)	7.0 (9)	1.4 (7)
	2. High blood pressure or hypertension	8.4 (37)	15.1 (18)	20.1 (27)	9.7 (49)
	3. Hyperlipidemia	4.3 (19)	6.7 (8)	3.0 (4)	0.4 (2)
	4. Stroke, small stroke, or TIA	0.2 (1)	0.8 (1)	1.5 (2)	0.0 (0)
	5. Asthma	0.7 (3)	0.0 (0)	0.0 (0)	0.6 (3)
	6. Diabetes	3.9 (17)	6.7 (8)	7.5 (10)	3.0 (15)
	7. Cancer	0.2 (1)	0.0 (0)	0.0 (0)	0.0 (0)
	8. Depression or Neurosis	0.5 (2)	0.8 (1)	0.7 (1)	0.0 (0)
	9. Other	8.7 (38)	12.6 (15)	8.2 (11)	0.8 (4)
Salt intake	Very high	4.6 (22)	5.0 (7)		3.1 (13)
Awareness of nutrition balance	No thought given to it	12.7 (61)	16.4 (23)		10.3 (51)
Alcohol intake	Almost everyday	1.6 (8)	1.4 (2)		1.6 (8)
Smoking habit	Smoking	13.6 (67)	15.3 (22)	15.3 (22)	18.6 (93)
Exercise frequency	Above 4 times a week	37.0 (182)	64.3 (92)	56.0 (79)	32.1 (161)
Body Mass Index (BMI)	Obesity (BMI > 29)	2.3 (11)	2.9 (4)	3.5 (3)	1.0 (5)

※ investigated in 1st survey.

**Table 10 ijerph-20-05450-t010:** Questions and evaluation scales for measuring moderating variables in the final version of the structural equation model.

Variables	Question	Scale
Frequency of opening bedroom windows	Do you open bedroom windows while sleeping in the dry/rainy season?	1: Rarely to 4: Always
Length of time at home	Thinking about the last twelve months (1st survey)/three months (2nd survey)/four months (3rd survey), how much of the day do you spend at home?	1: <8 h 2: 8–15 h 3: >15 h
Sleep disturbances	How often do you have any trouble getting to sleep or staying asleep?	1: Seldom 2: Sometimes 3: Often
Stress	Thinking about the amount of stress in your life, would you say that most days are stressful?	0: Not at all to 10: Extremely
Nutrition	Do you think about the nutritional balance of your diet?	1: Think a lot to 4: Do not think about it
Personal sensitivity	In daily life, how sensitive are you to the following environmental conditions: noise, coldness, odors, vibration	1: Not at all to 5: Extremely
Residential area preference and quality	Please evaluate your living area according to the following items: green space, street scenery, view from living areas.	1: Extremely good to 5: Extremely bad

**Table 11 ijerph-20-05450-t011:** Parameter estimates of the structural equation model for noise annoyance with opening of the bedroom window during the dry season.

Parameter	2019				2020			
	Estimate	SE	CR	*p*	Estimate	SE	CR	*p*
Living conditions ← *L*_den_	0.010	0.004	2.549	0.011	−0.005	0.004	−1.282	0.200
Sensitivity ← Living conditions	0.555	0.149	3.724	*	0.070	0.038	1.838	0.066
Health ← *L*_den_	−0.001	0.001	−0.912	0.362	−0.005	0.003	−2.093	0.036
Open window ← *L*_den_	−0.006	0.007	−0.851	0.395	−0.030	0.005	−5.490	*
Health ← Living conditions	0.051	0.055	0.942	0.346	0.007	0.027	0.253	0.801
Health ← Sensitivity	0.155	0.016	0.931	0.352	0.180	0.087	2.058	0.040
Annoyance ← Open window	−0.127	0.104	−1.222	0.222	−0.759	0.166	−4.565	*
Annoyance ← Health	−79.810	88.806	−0.899	0.369	5.677	2.666	2.130	0.033
Annoyance ← Sensitivity	1.863	0.614	3.033	0.002	−0.382	0.619	−0.618	0.537
Annoyance ← Living conditions	3.861	1.743	2.215	0.027	0.063	0.279	0.225	0.822

* *p* < 0.001; SE, standard error; CR, critical ratio (CR = estimate/SE).

**Table 12 ijerph-20-05450-t012:** Parameter estimates of the structural equation model for noise annoyance with length of time at home.

Parameter	2019 Survey				2020 Survey			
	Estimate	SE	CR	*p*	Estimate	SE	CR	*p*
Living conditions ← *L*_den_	0.010	0.004	2.568	0.010	−0.005	0.004	−1.267	0.205
Sensitivity ← Living conditions	0.579	0.150	3.823	*	0.092	0.042	2.174	0.030
Health ← *L*_den_	−0.009	0.005	−1.674	0.094	0.011	0.006	1.775	0.076
Length of time at home ← *L*_den_	−0.009	0.005	−1.663	0.096	0.018	0.004	4.725	*
Health ← Living conditions	0.246	0.143	1.722	0.085	0.003	0.053	0.050	0.960
Health ← Sensitivity	0.098	0.057	1.722	0.085	−0.308	0.183	−1.680	0.093
Annoyance ← Length of time at home	0.086	0.135	0.635	0.525	0.793	0.241	3.287	0.001
Annoyance ← Health	−11.995	7.472	−1.605	0.108	−2.397	1.210	−1.981	0.048
Annoyance ← Sensitivity	1.915	0.519	3.688	*	−0.128	0.571	−0.224	0.823
Annoyance ← Living conditions	2.719	1.136	2.393	0.017	−0.068	0.271	−0.250	0.803

* *p* < 0.001; SE, standard error; CR, critical ratio (CR = estimate/SE).

**Table 13 ijerph-20-05450-t013:** Parameter estimates of the structural equation model for insomnia with the opening of the bedroom window during the dry season.

Parameter	2019 Survey				2020 Survey			
	Estimate	SE	CR	*p*	Estimate	SE	CR	*p*
Living conditions ← *L*_night_	0.012	0.005	2.484	0.013	−0.014	0.005	2.857	0.004
Sensitivity ← Living conditions	0.291	0.068	4.307	*	0.075	0.050	1.495	0.135
Health ← *L*_night_	0.000	0.001	0.404	0.686	0.007	0.003	2.307	0.021
Open window ← *L*_night_	−0.014	0.007	1.872	0.061	−0.029	0.006	5.231	*
Health ← Living conditions	−0.007	0.018	0.405	0.686	0.059	0.032	1.812	0.070
Health ← Sensitivity	−0.010	0.026	0.406	0.685	−0.152	0.066	2.302	0.021
Insomnia ← Open window	−0.043	0.019	2.236	0.025	−0.006	0.029	0.222	0.825
Insomnia ← Health	−13.148	32.090	0.410	0.682	0.286	0.101	2.844	0.004
Insomnia ← Sensitivity	0.052	0.053	0.977	0.328	1.347	0.647	2.082	0.037
Insomnia ← Living conditions	0.027	0.041	0.652	0.514	−0.082	0.050	1.642	0.100

* *p* < 0.001; SE, standard error; CR, critical ratio (CR = estimate/SE).

**Table 14 ijerph-20-05450-t014:** Parameter estimates of the structural equation model for insomnia with length of time at home.

Parameter	2019				2020			
	Estimate	SE	CR	*p*	Estimate	SE	CR	*p*
Living conditions ← *L*_night_	0.012	0.005	2.487	0.013	−0.014	0.005	−2.857	0.004
Sensitivity ← Living conditions	0.291	0.067	4.307	*	0.075	0.050	1.494	0.135
Health ← *L*_night_	0.000	0.001	0.427	0.669	0.007	0.003	2.282	0.023
Length of time at home ← *L*_night_	−0.013	0.006	−2.234	0.025	0.017	0.004	4.839	*
Health ← Living conditions	−0.008	0.018	−0.429	0.668	0.058	0.032	1.799	0.072
Health ← Sensitivity	−0.011	0.025	−0.430	0.667	−0.151	0.066	−2.278	0.023
Insomnia ← Length of time at home	0.020	0.025	0.820	0.412	0.014	0.045	0.320	0.749
Insomnia ← Health	−12.162	27.965	−0.435	0.664	0.285	0.100	2.838	0.005
Insomnia ← Sensitivity	0.054	0.053	1.008	0.313	1.347	0.655	2.057	0.040
Insomnia ← Living conditions	0.047	0.041	1.136	0.256	−0.083	0.050	−1.661	0.097

* *p* < 0.001; SE, standard error; CR, critical ratio (CR = estimate/SE).

## Data Availability

Data are available from the authors upon reasonable request.

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
