# Peer review of "Models of Aviation Noise Impact in the Context of Operation Decrease at Tan Son Nhat Airport†"

_ijerph, 2023, doi:10.3390/ijerph20085450_

Round 1
Reviewer 1 Report
The article is of good scientific quality, written clearly in good English, and uses relevant scientific literature. However, there are some comments and questions about this article:
11. Discussion section is too short compared to the long result section, in lines, 469 – 471 authors did not cite similar studies on the mental health burden of the global pandemic of coronavirus. There were some studies published in Europe on traffic noise exposure decrease due to the coronavirus pandemic, but authors did not mention these studies (e.g. Wojciechowska W et al. Blood pressure and arterial stiffness in association with aircraft noise exposure: long- term observation and the potential effect of COVID-19 lockdown. Hypertension.
2021;78:325–334.., Hornberg, J. et al. Impact of the COVID-19 Lockdown Measures on Noise Levels in Urban Areas-A Pre/during Comparison of Long-Term Sound Pressure Measurements in the Ruhr Area, Germany. International Journal of Environmental Research and Public Health, April 2021, Volume 18: 9, 4653. Proceedings Internoise 2022)
22. In the discussion authors did not mention the strengths and limitations of the study.
33. The low response rate and the small sample size in the second survey require an explanation in the discussion section.
44. Authors did not explain how they administered the questionnaires and how were the questionnaires validated.
Reviewer 2 Report
Formally the article is well written. The analyzes are interesting. What puzzles me is that the factor "pandemic" is totally excluded, i.e. the factor "pandemic induced stress" is not taken into account or is not implemented in the model. This also coincides with the fact that insomnia is not only associated with noise, but also with the state of health. During the pandemic, concern about the global situation may have replaced the stress related to noise exposure, causing the insomnia data to remain unchanged. I would suggest putting remarks like this into the discussion.
Author Response
Thank you for your thoughtful feedback on the article. We appreciate your comment regarding the exclusion of the factor "pandemic induced stress" and how it may have impacted the data on insomnia. You make an interesting point about how the concern for the global situation during the pandemic may have replaced stress related to noise exposure as a cause of insomnia. We included your opinion in the discussion section in lines 634-644. Thank you again for your valuable input.
Reviewer 3 Report
Dear Authors,
The article is interesting and complies with the scientific profile of the 'International Journal of Environmental Research and Public Health'. The analyzes presented in the article are interesting, but I have the impression that (at least some of them) require a comment.
The work is of a research nature, concerns audible noise, it summarizes the results of surveys of residents in the vicinity of airports located on different sides of cities. The structural equation model was used for the calculations, taking into account the correlation between nonacoustic and acoustic factors.The article has been edited with great care. In my opinion - the simulation results are concluded correctly, just as the inference was correctly carried out.
SI units were used correctly in the body of the article. I consider all illustrations to be necessary for the content of the work.
General remarks:
1. In the reviewed work, I did not notice any attention to infrasound, which has a particular impact on insomnia. It would be useful to refer to publications on the impact of this type of impact and potential sources of formation in the phase of take-off and landing of aircraft.
2. The authors do not specify the type of sound level correction determined by the model, and more precisely with which correction curves the sound levels are calculated: A (adapts the measurement characteristics of the device to the sensitivity of the ear in the range of low volume levels), or D for aircraft noise.
3. The conclusions are strictly general. In my opinion, a few detailed conclusions should be added, based on the statements presented in the work.
Little things
4. Table 3. Average noise levels - there no units (dB).
5. Table 7. Non-acoustic components: Sensitivity (% Very and extremely) - a definition of determining the sensitivity parameter would be useful.
6. Please indicate the software used in simulation tests.
All the best,
Reviewer.
Reviewer 4 Report
General comments:
The paper describes a repeated survey conducted near a busy international airport, seeking to understand how the changes in air traffic caused by the pandemic influenced the relationship between acoustic and non-acosutic factors in the survey population. Detailed air traffic noise exposure was assessed, but the study does not address or assess potential and confounding exposures to other sources. This should at least be noted as a potential confounder if other sources are present, and if not this should be clarified as not of concern. Structural equation models are used to assess the relationships between noise, living conditions and personal characteristics such as health and sensitivity and how they influence annoyance and insomnia potentially caused by air traffic noise. The findings suggest notable and interesting changes in these relationships following altered noise patterns caused by pandemic shutdowns. Overall, the study is interesting and contributes an important observation and hypothesis to the literature, namely the structural relationship between noise effects on health and annoyance. The finding that annoyance is indirectly caused by perceptions of the living environment and more importantly health status is intriguing and should be promoted more centrally as an outcome of the paper. This relates to the general criticism of the work that there did not appear to be a strong hypothesis for setting up the SEMs, although there clearly were some ideas put into the initial structures. They authors would therefore be well advised to provide a bit more information about the theoretical framework that was used to build the SEM structure in the first place, and consequently discuss how their findings deviate from this as the discussion is generally quite sparse. To this end, the very title of the paper is somewhat misleading and does not represent what is actually discovered in the research!
Specific comments:
- The Abstract has several writing errors, missing “and” on line 28
- Use of Lden range to describe noise levels in abstract and introduction is not informative, should at least include mean and standard deviation in addition to max/min so reader gets an idea of the distribution, but furthermore the noise levels do not provide any information about the level of disturbance so should instead report how many people are within noise mapping area (presumably with airport noise levels above 50 dB) or number of people above a given threshold level
- Section 2.3 – More details on noise model, what model specifically, software, building reflections if applicable, calculation window (e,g, hourly?)
- Line 165-176: Values from 10-29% of survey population above 60 corresponds to census? Statement is not necessary as values are reported in table..
- Line 183 unclear
- Line 195-199: Removing from analysis might be justifiable, but could be related to increased stress caused by the pandemic in general, and how do you know that this is not impacting sleep for the survey populations in general, potentially biasing the SEMs? Should be addressed in discussion
- Tables 4-6 are somewhat confusing, why not report for all respondents in each survey? Not only respondents who continued participation are included in the multivariate analyses
- Where did questions on environmental sensitivities and ‘Residential area preference and quality’ come from? Not described in methods. Sensitivity questions are fairly self-explanatory in terms of what was asked, but not clear what for example Green, Street sceneries and view from houses refers too, perhaps reword the short question descriptions in the table or add a paragraph in methods for these two groups of questions. Some information is given in Table 8 but should not have to search for this information at a later point after presented in Table 7
- Line 260-261: What was the ‘modification process’ used? Trial and error to find an acceptable SEM with common variables for each survey?
- Table and figure headings should be more descriptive in general (e.g., table 8, what model? Are these all the variables used in the SEMs)
-
